

# Antibodies to *Bordetella pertussis* antigens in maternal and cord blood pairs: a Thai cohort study

Nasamon Wanlapakorn[1], Thanunrat Thongmee[1], Preeyaporn Vichaiwattana[1], Elke Leuridan[2], Sompong Vongpunsawad[1] and Yong Poovorawan[1]

[1] Center of Excellence in Clinical Virology, Department of Pediatrics, Faculty of Medicine, Chulalongkorn University, Bangkok, Thailand
[2] Centre for the Evaluation of Vaccination, Vaccine & Infectious Disease Institute, University of Antwerp, Belgium

## ABSTRACT

**Background**. Pertussis is a vaccine-preventable disease, yet an increasing incidence of pertussis occurs in many countries. Thailand has a long-standing pertussis vaccination policy, therefore most expectant mothers today had received vaccines as children. The resurgence of pertussis among Thai infants in recent years led us to examine the pre-existing antibodies to *Bordetella pertussis* antigens in a cohort of 90 pregnant women.
**Methods**. We evaluated the IgG to the Pertussis toxin (PT), filamentous hemagglutinin (FHA) and pertactin (PRN) in maternal and cord blood sera using commercial enzyme-linked immunosorbent assays (ELISA).
**Results**. When values of >10 IU/ml were accepted as potential protective concentrations, we found that the percentages of unprotected infants were 73.3%, 43.3% and 75.5% for anti-PT, anti-FHA and anti-PRN IgG, respectively.
**Discussion**. These results may explain the susceptibility for pertussis among newborn infants in Thailand and support the requirement for a pertussis booster vaccine during pregnancy, which may contribute to the passive seroprotection among newborns during the first months of life.

## INTRODUCTION

Countries with universal pertussis vaccination have experienced pertussis resurgence in recent years especially those implementing the acellular pertussis (aP) vaccines (*Clarke et al., 2013*; *Octavia et al., 2012*; *Van der Maas et al., 2013*; *Winter et al., 2014*). Recent evidence suggests that this was partly due to the waning immunity as a result of the aP-vaccine use (*Klein et al., 2012*; *Liko, Robison & Cieslak, 2013*). Maternal vaccination during pregnancy is an effective strategy to prevent pertussis-related morbidity and mortality in newborns who are at the highest risk for infection and hospitalization (*ACIP, 2013*). Vaccination during pregnancy boosts the immune response against *B. pertussis* in expectant mothers and affords the transplacental transfer of antibodies to the baby, thus conferring protection to pertussis in infants during the first few months of life (*Hoang et al., 2016*; *Maertens et al., 2016*; *Munoz et al., 2014*; *Vizzotti et al., 2016*).

Corresponding author
Yong Poovorawan,
Yong.P@chula.ac.th

Thailand implemented a routine infant immunization program with two doses of the Diphtheria–Tetanus toxoid–whole-cell Pertussis (DTwP) vaccine beginning in 1977. From 1992 onward, the country's Expanded Program on Immunization (EPI) offers five doses of DTwP to infants at two, four, six, 18 and 48 months of age. A booster dose of Tetanus Toxoid (TT), which has since been replaced by diphtheria–tetanus vaccine (dT) in 2012, is also recommended during adolescence and every 10 years thereafter. Since then, data from the national passive surveillance showed a relatively low burden of pertussis in the general population. Between 2007 and 2014, the incidence of pertussis in Thailand was reportedly 6–25 per year (0.01–0.04 per 100,000 individuals) (*Thailand Bureau of Epidemiology, 2014*). In 2015 and 2016, however, there was an increased incidence of 51 cases (0.08/100,000) and 72 cases (0.11/100,000), respectively (*Thailand Bureau of Epidemiology, 2016*). Significant numbers of morbidity belonged to children one year of age or younger and this age group bears the greatest risk of pertussis morbidity. In addition, these numbers are likely to be under-estimated due to missed pertussis diagnosis and inadequate laboratory confirmation. To date, Thailand has not yet integrated the Tetanus-reduced dose of diphtheria and acellular pertussis (Tdap) booster during adolescence or pregnancy.

Despite the universal whole-cell pertussis (wP) vaccination among newborns in Thailand, anti-PT IgG has been shown to wane precipitously (*Wanlapakorn et al., 2016*). To evaluate the susceptibility to pertussis among infants born to Thai mothers, we aimed to determine the baseline concentration of IgG against PT, FHA and PRN in pregnant women who did not receive pertussis vaccination during pregnancy. Data from this study may be important in providing evidence-based consideration for a pertussis booster during pregnancy.

## MATERIALS AND METHODS

### Study population

The study was approved by the Institutional Review Board of the Faculty of Medicine of Chulalongkorn University (IRB No. 154/58). The serum samples were archived residual samples from 90 mother-cord blood pairs collected between July 2011 and August 2012 to examine serological protection against tetanus among pregnant women at King Chulalongkorn Memorial Hospital in Bangkok. All samples were deidentified and anonymous, therefore no consent was required and the permission to use these samples was granted by the Director of King Chulalongkorn Memorial Hospital. Inclusion criteria were healthy pregnant Thai women between 15 and 45 years who sought antenatal care at King Chulalongkorn Memorial Hospital in 2011 and 2012. During their first visit, history of tetanus immunization and blood samples were obtained. Cord blood samples were collected at the time of delivery. None of the mothers in this study were able to provide their vaccination records, therefore women born prior to 1977 were assumed to have never received DTwP, while those born after 1977 were assumed to have had between two and five doses of DTwP.

## Antibody measurement

The anti-PT, anti-FHA and anti-PRN IgG were analyzed quantitatively using commercial ELISA kits (EUROIMMUN, Lübeck, Germany) according to the manufacturer's instructions. The controls of the ELISA kits were calibrated using the first International World Health Organization standards (WHO International Standard Pertussis Antiserum, human, 1st IS NIBSC Code 06/140) and quantified in international unit per milliliter (IU/ml). The international reference preparation of the Food and Drug Administration (Bethesda, MD, USA) was used. Sera were initially diluted 1:101 for the test and higher dilutions were performed as necessary. The lower limit of quantification (LLOQ) for anti-PT, anti-FHA and anti-PRN IgG is 5 International Unit (IU) per ml of serum. Values below LLOQ were calculated as half of the LLOQ. For anti-PT, Values <5 IU/ml were interpreted as seronegative, 5–40 IU/ml as no evidence of recent acute infection, 40–100 IU/ml as probable past exposure to pertussis, and >100 IU/ml as acute pertussis infection or recent vaccination.

## Statistical analysis

The IgG level was expressed as geometric mean concentrations (GMC) with standard error of the mean (SE). Data were analyzed using SPSS software (IBM Inc., Armonk, NY, USA), SigmaPlot (Systat Software, San Jose, CA, USA) and R statistical software. Chi square and Fisher's exact test were used for statistical comparisons of seronegativity rates of pregnant women born before or after the pertussis inclusion in the EPI. Linear regression model was used to predict antibody levels in cord sera. A simple and multivariable regression models were used to analyze the predictors affecting the antibody level in the cord blood.

## RESULTS

Recent statistics in Thailand showed an increase in the number of pertussis in the 0–1 age group (Table 1). Between 2013 and 2015, 3 pertussis-related deaths were reported. To determine whether expectant mothers in recent years possessed any immunity against *B. pertussis*, we tested a cohort of convenient serum samples for anti-PT, anti-FHA, and anti-PRN IgG obtained from mothers and the cord blood, the latter of which served as a surrogate for infant blood samples at birth. The majority of maternal blood samples were obtained during the first trimester of pregnancy as defined by the gestational age (GA) at less than 12 weeks (Table 2). All babies except one were born healthy. One baby born at GA of 23 weeks died hours after birth due to multiple congenital anomalies.

The GMC of anti-PT, anti-FHA and anti-PRN IgG in maternal and cord sera (Fig. 1) were derived from individual values (Table S1). Maternal and cord sera samples demonstrated similar anti-PT and anti-PRN levels, both of which were lower than that of anti-FHA. When values of >10 IU/ml were accepted as potential protective concentrations, we found that the percentages of unprotected infants were 73.3%, 43.3% and 75.5% for anti-PT, anti-FHA and anti-PRN IgG, respectively. Comparison of all 90 mother-infant paired samples showed that maternal anti-PT, anti-FHA and anti-PRN IgG correlated significantly with cord blood values (Fig. 2).

**Table 1  Pertussis incidence in Thailand between 2011 and 2016 by age group.** Data were retrieved from the annual epidemiology surveillance report by the Bureau of Epidemiology, Department of Disease Control, Ministry of Public Health, Thailand. The numbers represented suspected, probable and confirmed cases[a] reported annually.

| Age group (years)<br>Year | 0–1 | 1–4 | 5–9 | 10–14 | 15–24 | 25–34 | 35–44 | 45–54 | 55–64 | 65 and above |
|---|---|---|---|---|---|---|---|---|---|---|
| 2016 | 27 | 16 | 9 | 4 | 3 | 5 | 3 | 0 | 0 | 5 |
| 2015 | 35 | 8 | 3 | 0 | 0 | 1 | 0 | 1 | 2 | 1 |
| 2014 | 11 | 1 | 1 | 1 | 0 | 1 | 0 | 0 | 0 | 1 |
| 2013 | 11 | 4 | 1 | 1 | 0 | 1 | 2 | 2 | 1 | 1 |
| 2012 | 6 | 2 | 3 | 3 | 1 | 0 | 1 | 1 | 0 | 0 |
| 2011 | 7 | 2 | 1 | 1 | 0 | 0 | 1 | 0 | 0 | 0 |

Notes.

[a] Pertussis case definitions; A suspected case is a patient presenting with a cough illness lasting ≥2 weeks with at least one of the following signs or symptoms: paroxysms of coughing; or inspiratory "whoop"; or post-tussive vomiting. A probable case is defined as a suspected case with epidemiologic linkage to a laboratory-confirmed case. A confirmed case is defined as a suspected case with laboratory confirmation by polymerase chain reaction or bacterial culture.

**Table 2  Demographic characteristics of pregnant women and infants in the study.** Data were presented as mean and range.

| Characteristics | |
|---|---|
| Nunmber of pregnant women | 90 |
| Mean age in years (range) | 30.9 (19–42) |
| No. of participants whose maternal sera were collected during | |
| - First trimester (GA ≤ 12 weeks) | 65 (72%) |
| - Second trimester (GA13–28 weeks) | 24 (27%) |
| - Third trimester (GA ≥ 29 weeks) | 1 (1%) |
| Mean GA at maternal blood collection in weeks (range) | 11 (5–29) |
| Mean GA at delivery in weeks (range) | 37.6 (23–40) |
| Infant birth weight in grams (range) | 2,952 (980–4,060) |
| Percentage of premature delivery (GA < 37 weeks) | 11.1% |
| Percentage of twins | 3.3% |

Anti-PT IgG levels can reflect not only the circulating antibody from past vaccination but also the recent exposure to pertussis. When the levels were classified as <5 IU/ml (seronegative), 5–40 IU/ml (no evidence of recent infection), 40–100 IU/ml (probable past exposure to pertussis) and >100 IU/ml (acute or recent infection). We found that 56.7% of pregnant women were seronegative for anti-PT. Forty percent of the maternal serum samples did not show evidence of recent infection, as defined by antibody levels between 5–40 IU/ml. Three women (aged 19, 28, and 40 years) possessed anti-PT IgG titers of >100 IU/ml, which represented 3.3% of the samples. Two-thirds of these high anti-PT IgG samples also possessed >350 IU/ml of anti-PRN IgG, while one-third had >350 IU/ml of anti-FHA IgG, suggesting possible recent infection.

Pregnant women who were born before the pertussis vaccine was implemented into the EPI program (30/90) were 34 years of age or older (Table 3). There were no differences

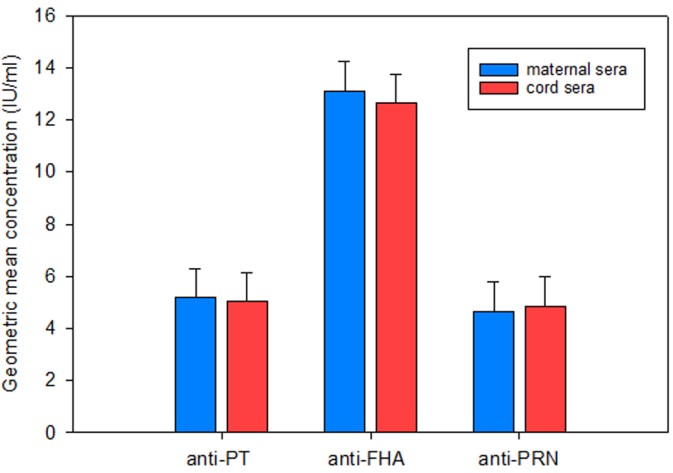

**Figure 1 Maternal and cord anti-PT, anti-FHA and anti-PRN IgG in this study.** Values are expressed as geometric mean concentration (GMC) in IU/ml. Error bars indicated the standard error (SE) of the mean. Mean + SE (IU/ml) for each concentration were: anti-PT in maternal sera (5.19 + 1.11), anti-PT in cord sera (5.05 + 1.11), anti-FHA in maternal sera (13.10 + 1.13), anti-FHA in cord sera (12.64 + 1.12), anti-PRN in maternal sera (4.65 + 1.12), and anti-PRN in cord sera (4.84 + 1.12).

**Table 3 Anti-PT IgG in pregnant women born before or after the implementation of pertussis vaccination.** There were no differences in the proportion of seronegativity or recently infected rates in women born before or after the EPI program.

| Anti-PT IgG (IU/ml) | Before EPI (aged ≥ 34 years) ($n = 30$) | After EPI (aged < 34 years) ($n = 60$) | $p$-value |
|---|---|---|---|
| <5 | 16 | 35 | 0.660 |
| 5–40 | 13 | 23 | 0.656 |
| 40–100 | 0 | 0 | – |
| >100 | 1 | 2 | N/A |

**Notes.**
N/A, Not Applicable.

in the proportion of anti-PT IgG seronegativity or recently infected rates in women born before or after the EPI program.

To determine factors affecting the antibody level in the cord sera, we tested two models. Using the bivariable model, which assumed that level of anti-PT, anti-FHA and anti-PRN in cord sera was entirely attributed to the maternal sera, we found that antibody level in maternal sera significantly affected the level in cord sera ($p < 0.001$). This model would account for 88.4%, 89.2% and 95.9% of the variations in cord anti-PT, andti-FHA and anti-PRN IgG, respectively. Using the multivariable model, which took the gestational age at delivery and the interval between maternal and cord sera collection into consideration, neither parameters were significantly associated with the level in cord sera and did not confound the effect of maternal sera.
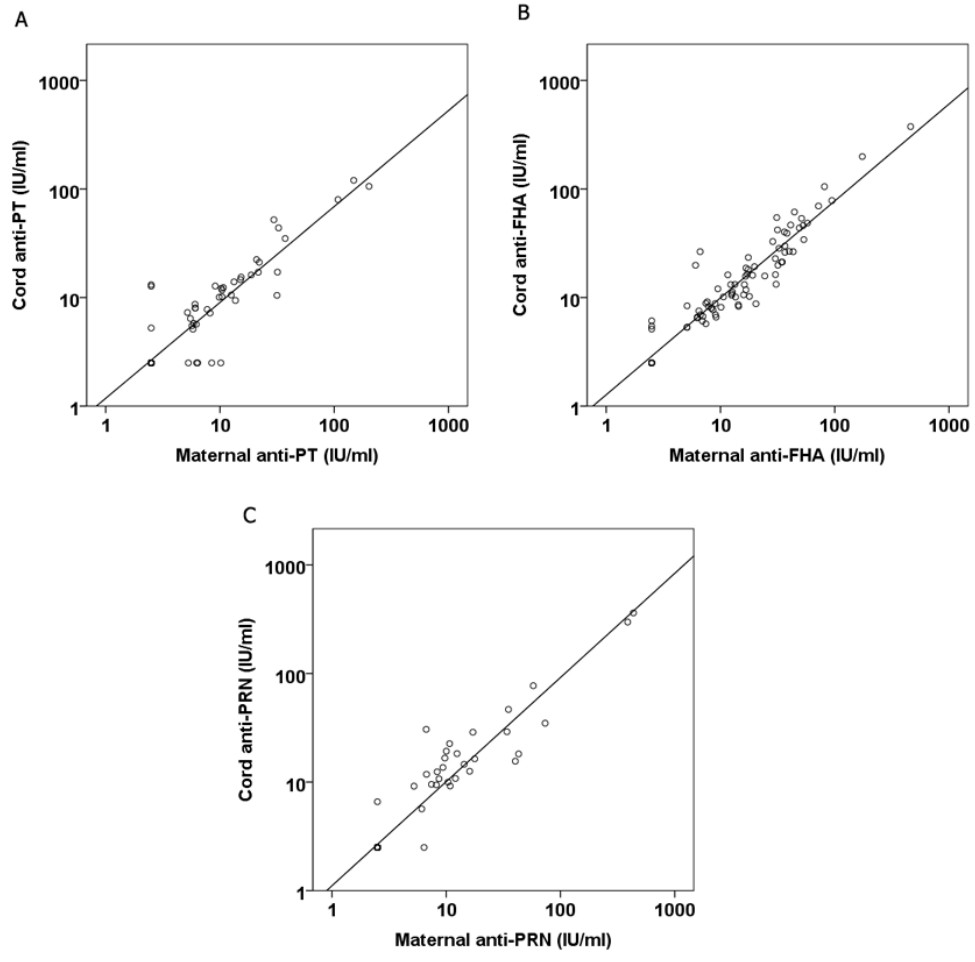

**Figure 2** **Simple linear regression model of IgG to PT (A), FHA (B) and PRN (C) in maternal and cord sera.** $R^2$ for anti-PT = 0.854, $p$ value < 0.001, $R^2$ for anti-FHA = 0.894, $p$ value < 0.001, $R^2$ for anti-PRN = 0.913, $p$-value < 0.001.

## DISCUSSION

The increase in the prevalence of reported pertussis morbidity among infants in Thailand in recent years led us to examine the prevalence of IgG to PT, FHA and PRN in maternal and cord blood paired samples in order to determine serological baseline. Despite receiving wP vaccination as children, most pregnant women in this study demonstrated low anti-PT, anti-FHA and anti-PRN IgG, which were also reflected by the serology status in the corresponding cord serum samples. The finding suggests that a significant number of women and their newborn children are susceptible to pertussis. Additionally, the anti-PT seronegative rates were similar irrespective of vaccination status and implicated waning immunity for individuals who had received wP vaccination.

Previous studies examining seroprevalence of anti-PT IgG in pregnant women revealed that maternal GMCs of anti-PT detected at delivery ranged between 2.4–19.39 ELISA Units or IU/ml (*De Voer et al., 2009*; *Ercan et al., 2013*; *Gonik et al., 2005*; *Healy et al., 2004*; *Healy*

*et al., 2006; Heininger et al., 2009; Hoang et al., 2016; Shakib et al., 2010; Van den Berg et al., 2010; Villarreal Perez et al., 2017*). Although our GMC of 5.18 IU/ml was towards the low side of this range, it was similar to the values reported from the nationwide seroprevalence survey of anti-PT IgG in 2014 (*Wanlapakorn et al., 2016*). In that study, the GMCs of anti-PT IgG were 4.53 and 3.76 IU/ml among 21–30 and 31–40 year-olds, respectively. This reflects the susceptibility to pertussis among women of child-bearing age in Thailand and supports the potential benefit of a booster during adolescence and pregnancy.

FHA is a cell surface protein found in many bacteria including *Bordetella pertussis*. It plays a key role in bacterial adhesion to epithelial cells and thus contributes to its pathogenesis. Previous studies have shown that women who did not receive pertussis vaccination during pregnancy had anti-FHA IgG ranging from 6.9 to 26.6 ELISA units or IU/ml (*De Voer et al., 2009; Ercan et al., 2013; Gonik et al., 2005; Healy et al., 2004; Heininger et al., 2009; Hoang et al., 2016; Van den Berg et al., 2010*). Our results were consistent with these reported values. In addition to PT, PRN is another virulence factor, which promotes bacterial adhesion to epithelial cells. Anti-PRN concentrations correlated with protection to pertussis disease as they facilitate phagocytosis by polymorphonuclear cells (*Hellwig et al., 2003*). The levels of anti-PRN IgG in this study were consistent with those reported in recent studies, which found that anti-PRN IgG in maternal sera were between 4.09–13.5 ELISA units or IU/ml (*De Voer et al., 2009; Gonik et al., 2005; Hoang et al., 2016; Van den Berg et al., 2010; Villarreal Perez et al., 2017*).

It has been reported that approximately 1.8–6.3% of pregnant women possessed anti-PT IgG levels of >100 IU/ml, which indicated recent *B. pertussis* infection (*Nooitgedagt et al., 2009; Plans et al., 2014; Plans et al., 2008*). These women were presumed to be protected from pertussis during pregnancy and were shown to transfer anti-PT IgG to their fetus. With the assumption of vaccination based on age, but without available vaccination record or clinical confirmation of pertussis, we concluded that 3.3% of mothers with anti-PT IgG titer >100 IU/ml in this study had recently been exposed to *B. pertussis*. For 30–40% of pregnant women who had anti-PT IgG titers between 5–40 IU/ml, we cannot eliminate the possibility of waning immunity associated with *B. pertussis* exposure sometime in the past few years. If this was indeed representative of the whole population, then it is plausible that *B. pertussis* exposure is not uncommon despite universal childhood vaccination in Thailand.

Previous studies suggested that transplacental transfer of antibodies occurs via active transport, which result in the increased level of antibodies in the cord blood at term when compared to maternal sera (*Ercan et al., 2013; Hardy-Fairbanks et al., 2013; Heininger et al., 2009*). During early infancy, therefore, protection from diseases often relies on the passively transferred maternal antibodies to newborn infants. In this study, we demonstrated that maternal anti-PT, anti-FHA and anti-PRN IgG levels highly correlated with those of the paired cord blood. We demonstrated that the majority of Thai infants were unlikely to be protected from pertussis regardless of the mothers' vaccination status. Although the increase in reported pertussis incidence seen in very young children may reflect better disease awareness by clinicians, it may also reflect the under-diagnosis of pertussis in adults and the waning of the immunity to *B. pertussis* in the general population.

Future work to assess the effect of a booster dose of pertussis vaccination among pregnant women in terms of antibody to *B. pertussis* antigens, the extent of passive maternal-fetal antibody transfer, and long term immunity in Thai infants induced by whole cell or acellular pertussis vaccine is ongoing. To reduce the risk of pertussis-related complications and infant mortality in countries using wP vaccine in their EPI program, further studies to examine the effectiveness of Tdap vaccination during pregnancy, the long-term effect of Tdap in wP-vaccinated infants, and possible interference of maternal-derived antibody with wP-vaccinated infants will be important.

## CONCLUSIONS

Our results explained the susceptibility for pertussis among newborn infants in Thailand and supported the requirement for a pertussis booster vaccine during pregnancy. Maternal vaccination during pregnancy may provide passive seroprotection in newborns during the first months of life.

## ACKNOWLEDGEMENTS

The authors would like to thank Professor Teerapong Tantawichien in the Department of Medicine (Faculty of Medicine, Chulalongkorn University) for providing serum samples from the project that aimed to study tetanus serological protection among pregnant woman at King Chulalongkorn Memorial Hospital. We also would like to thank the staff at the Center of Excellence in Clinical Virology and King Chulalongkorn Memorial Hospital for technical assistance.

### Funding

This work was supported by the Research Chair Grant NSTDA (P-15-50004), the Integrated Innovation Academic Center and the Chulalongkorn University Centenary Academic Development Project (CU56-HR01) of Chulalongkorn University, the Ratchadaphiseksomphot Endowment Fund of the Center of Excellence in Clinical Virology (GLE 58-014-30-004, RES560530093), the National Research University Project, Office of Higher Education Commission (NRU59-002-HR) and Siam Cement Group, MK Restaurant Company Limited. The funders had no role in study design, data collection and analysis, decision to publish, or preparation of the manuscript.

### Grant Disclosures

The following grant information was disclosed by the authors:
Research Chair Grant NSTDA: P-15-50004.
Innovation Academic Center and the Chulalongkorn University Centenary Academic Development Project: CU56-HR01.
Center of Excellence in Clinical Virology: GLE 58-014-30-004, RES560530093.
National Research University Project, Office of Higher Education Commission: NRU59-002-HR.

Siam Cement Group.

## Competing Interests

The authors declare there are no competing interests.

## Author Contributions

- Nasamon Wanlapakorn conceived and designed the experiments, performed the experiments, analyzed the data, wrote the paper, prepared figures and/or tables, reviewed drafts of the paper.
- Thanunrat Thongmee and Preeyaporn Vichaiwattana performed the experiments, contributed reagents/materials/analysis tools.
- Elke Leuridan analyzed the data, wrote the paper, reviewed drafts of the paper.
- Sompong Vongpunsawad analyzed the data, contributed reagents/materials/analysis tools, wrote the paper.
- Yong Poovorawan conceived and designed the experiments, analyzed the data, wrote the paper, provide funding.

## Human Ethics

The following information was supplied relating to ethical approvals (i.e., approving body and any reference numbers):

The Institutional Review Board of the Faculty of Medicine, Chulalongkorn University, Bangkok, Thailand.

## Data Availability

The raw data has been provided in a Supplemental File.

## Supplemental Information

Supplemental information for this article can be found online at http://dx.doi.org/10.7717/peerj.4043#supplemental-information.

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
