# Peer review of "Antibodies to Bordetella pertussis antigens in maternal and cord blood pairs: a Thai cohort study"

_PeerJ, doi:10.7717/peerj.4043_

## Round 0.1 · original submission · Minor Revisions

Dear Authors,

The Reviewers are favorable to the publication of your manuscript in PeerJ after a minor revision.

Please incorporate or discuss the suggested changes and submit a revised version of your manuscript in order to achieve publication.

Best regards

Salvatore Andrea Mastrolia
PeerJ Academic Editor

Reviewer 1 ·

Basic reporting

The article meets all basic reporting criteria.

Experimental design

The work is original and the study design is appropriate and clearly explained. The methods are well described and the analyses and presentation of results are adequate.

Validity of the findings

The investigators establish the poor level of protection against pertussis in both mothers and infants in a Thai population, setting the stage for additional work in support of maternal immunization with Tdap. Interestingly, this work in a location where whole cell pertussis vaccine is given to infants provides an opportunity to understand the role of adolescent and maternal vaccination with acellular pertussis vaccine followed by whole cell vaccine in infants.
The discussion and conclusions of this study are well stated and supported by the data presented.

Additional comments

Well written manuscript, with relevant and up to date references, clear description of methods, analyses and results. The figures are illustrative of the key findings of the study.

Reviewer 2 ·

Basic reporting

Figure 1, showing the distribution of declared pertussis cases according to age from 2011 up to 2016 is not very informative, as only for 2015 (n=51) and 2016 (n=72) the figures can be split in reasonable numbers. I would suggest to put the data of Figure 1 in a Table and also indicate how many of the cases were laboratory defined and by what method.

Experimental design

No comment

Validity of the findings

One minor comment on the conclusions:
-Line 149 states: There were no statistically significant differences in the proportion of recently infected rates in women born before or after the EPI program: statistics on 1 vs 2 cases? This can not be used for a statistical analysis

Additional comments

A general comment:
Line 227: Our results explained the susceptibility for pertussis among newborn infants in Thailand and supported the requirement for a pertussis booster vaccine during pregnancy. It is clear that pertussis booster vaccination during pregnancy will increase the antibodies in newborns, but it is also clear there are no actual correlates of protection for pertussis. Although antibodies to PT and Prn certainly play a role in protection, there is also evidence that other factors such as local IgA levels and cell-mediated immune responses are important.

---

## Round 0.2 · accepted · Accept

Dear Authors,
I would like to compliment with you for the efforts provided in addressing the Reviewers' comments.

All Reviewers now fell that your manuscript has reached the level of publication and can be accepted in its current form.

Best regards

Salvatore Andrea Mastrolia
PeerJ Academic Editor

Reviewer 2 ·

Basic reporting

no comment

Experimental design

non comment

Validity of the findings

no comment

Additional comments

no comment